# Single Cell RNA Sequencing and Its Impact on Understanding Human Embryo Development

**DOI:** 10.3390/ijms26167741

**Published:** 2025-08-11

**Authors:** Samina Gul, Chengting Zhang

**Affiliations:** State Key Laboratory of Primate Biomedical Research, Kunming University of Science and Technology, Kunming 650500, China; saminagul@kust.edu.cn

**Keywords:** scRNA, embryogenesis, stem cell, human embryo

## Abstract

Human embryonic development represents a crucial period of cellular specification and tissue organization, laying the foundation for all subsequent growth and differentiation. Because of its ethical and technical limitations, scientists use rare embryo samples and new in vitro models, such as stem cell-derived embryo-like structures. Our knowledge of human embryonic development has been completely transformed by single-cell RNA sequencing. This review covers the subjects of human embryogenesis, limitations in embryo research, the emergence of cultured embryo models, and how scRNA-seq has ultimately shaped the future of human developmental biology by becoming essential for analyzing developmental processes and evaluating the accuracy of stem cell-derived models.

## 1. Introduction

The concept of life includes the potential of reproduction, inheritance, and development [1]. Embryonic development is a complex process controlled by a number of specific cellular behaviors as well as unique genetic predispositions, genes, and environmental influences [2].

Research on early human development is essential for advancing knowledge of our genetics and the origins of human life. These studies can also provide insight into the causes of congenital diseases, early miscarriages, and infertility. The initial phase of human embryonic development is a cornerstone of life sciences, offering crucial insights into the processes that transform a single fertilized cell into a complex multicellular organism. This complicated journey encompasses key events such as cell fate decisions, lineage specification, and tissue organization, which are essential to understanding normal development and the cause of developmental disorders [3,4]. However, access to early human embryos is restricted, and thorough research is hampered by the ethical, technical, and legal difficulties associated with studying human embryonic development.

Stem cell technologies, single-cell RNA sequencing, and in vitro models like blastoids and gastruloids have all contributed to the tremendous advancements in human embryo research in recent years, offering new insights into early lineage specification, signaling pathways, and spatial gene expression patterns during early human development.

There are still major gaps in our understanding; for instance, post-implantation development, gastrulation, and early organ formation in humans remain poorly understood. This is partially due to ethical, legal, and technical restrictions that make it difficult to access and study human embryos, particularly after 14 days of development (the commonly known “14-day rule”). These challenges have driven the development of alternative models like stem cell-derived embryo-like structures, which mimic certain features of natural embryos without using fertilized human eggs. Despite their promise, these models are not yet completely accurate representations of embryonic development, which makes it challenging to reach firm conclusions.

The most recent developments in human embryonic development research are the main topic of this study, which also identifies knowledge gaps. We talk about the function of stem cell-based models, like gastruloids and blastoids, in simulating human development. Furthermore, we investigate the use of single-cell RNA sequencing (scRNA-seq) as a potent technique for validating these models, offering important information on how well they mimic the developmental trajectories and patterns of gene expression of real human embryos. By integrating recent findings, we aim to assess the strengths and limitations of these in vitro systems and their potential to revolutionize human developmental biology.

## 2. Human Embryogenesis from Blastocyst to Gastrulation

Human embryonic development begins with the production of the totipotent zygote during fertilization. The zygote forms at the beginning of human embryonic development and divides to produce the embryo and extra-embryonic components. At about the 20-cell stage, the cavity formation initiates, and fluid is pumped between the embryo’s cells (Figure 1). The embryo has now progressed to the early blastocyst stage. The blastocyst has a population of inner cell mass (ICM) cells that are asymmetrically positioned in relation to the outer layer of trophectoderm (TE), which encloses the blastocyst cavity. The ICM is composed of two cell types: epiblast and hypoblast. The latter creates an epithelial layer on the epiblast’s luminal side. The epiblast progressively leaves its pluripotent state around E7–E9 (CS5a) and transforms into a collection of radially orientated cells encircling a tiny primordial cavity known as the pro-amniotic cavity. Under the trophectoderm (TE), primitive endoderm (PE) cells proliferate to produce parietal PE (parPE), which then organizes beneath the ventral side of the epiblast to form the ventral endoderm (VE). Epiblast cells on the dorsal portion of the pro-amniotic cavity create a layer of squamous amnion epithelium during E10–E11 (CS5b), when the amniotic cavity (AC) is formed through pro-amniotic cavity expansion. The cells on the ventral portion mature into late epiblast while maintaining their goblet shape. Distal PE cells and VE cells combine to produce the primary yolk sac at the same time. The epiblast enlarges into a symmetric plate shape at E12–E13 (CS5c). The secondary yolk sac consists of a plate-shaped structure on the ventral part of the epiblast called the ‘bilaminar disc’ formed from the rearrangement of VE [5,6,7].

The human embryo begins the process of gastrulation at E14 (CS6). Because it lays the groundwork for the three germ layers and the body plan, gastrulation is a significant occurrence in the early stages of embryonic development. A “gastrula” is an embryo that has three germ layers at the gastrulation stage. Based on the epithelial–mesenchymal transition of T (brachyury)-expressed gastrulating cells (Gast), the PS first appears at the posterior area of epiblasts during human embryonic gastrulation, which spans CS7–CS9. The PS serves as a reference for the convergence of epiblast cells and establishes the body’s midline [5,6,7]. Furthermore, the asymmetric patterning of epiblasts along the anteroposterior axis is shown by the appearance of Gast and PS. An excellent summary of the main anatomical characteristics of human middle and late gastrulation has been provided [8]. From peri-implantation to gastrulation, the biology of human embryo development is far from being uncovered. In embryogenesis, deciphering the temporal and spatial patterns of gene expression is a critical step toward understanding the early developmental process. In this section, we present studies that used single-cell analysis to increase our understanding of the transcriptional and epigenetic landscape for early embryogenesis, such as preimplantation, gastrulation, organogenesis, and in vitro models like blastoids and gastruloids.

## 3. Human Embryo Research Progress Through the Lens of Single-Cell RNA Sequencing

Single-cell sequencing (SCS) is a new technology (Figure 2A,B) that examines a single cell’s genome, transcriptome, or epigenetics to reveal the functions and characteristics of cells at different stages from different perspectives. Tang et al. [9] developed the first single-cell RNA sequencing (scRNA-seq) technique based on microarray analysis, and as scRNA-seq techniques have developed and improved, they have become increasingly popular in studies of embryonic development.

## 4. Preimplantation Development

The separation of three cell lineages and X chromosome inactivation (XCI), which needs to be resolved at single-cell resolution, are two examples of the many ways that human preimplantation development differs from that of mice. Because of a limited number of cells in the early stages of embryonic development, their properties cannot be systematically exploited. Nevertheless, the research landscape of transcriptomics is continuously changing due to the emergence of next-generation sequencing technologies, with scRNA-seq uncovering dynamic gene expression during early embryonic development, differentiation, and reprogramming [10]. Nearly 2000 individual cells were examined in a number of publications that presented single-cell RNA sequencing (scRNA-seq) data for human preimplantation embryos [3,11,12,13,14,15]. According to these findings, the maternal-to-zygotic transition (MZT) and the differentiation of the blastomeres into three cell lineages are reflected in the highly dynamic transcriptome of preimplantation embryos and the ability to discriminate between cells of different stages.

The expression of 124 genes from the preimplantation development of human embryos and human embryonic stem cells (hESCs) was analyzed by Yan et al. [15] using scRNA-seq. They were able to identify 22,687 expressed genes, including 8701 long non-coding RNAs, which is significantly more than the number of genes found by cDNA microarray. This study is noteworthy because it is the first to analyze the transcriptome of hESCs as early as passage 0, which allows for the tracking of the earliest changes in gene expression from epiblasts to hESCs. The scRNA-seq findings indicate that the most notable shift in gene expression takes place between the four- and eight-cell stages, which is consistent with the timing of the main MZT [15]. The zygotic-specific machinery for transcription, translation, epigenetics, and cell division is established as zygotic genome activation (ZGA), as evidenced by the approximately 2500 genes that are upregulated at that time with strong enrichment for the Gene Ontology (GO) terms “RNA metabolism and translation,” “chromosome organization,” “cell division,” and “DNA packaging” [15]. The late activation of the Y chromosomal genes at the morula stage is an example of how the ZGA is still incomplete at the eight-cell stage [3]. Petropoulos et al. [3] utilized single-cell sequencing to systematically demonstrate the transcriptome map of the pre-implantation development of human embryos, uncovering that cells are in an intermediate state of co-expression of chromosome-specific genes. Therefore, we may gain a deeper understanding of human development and embryonic stem cells. From the two-cell stage to the four-cell stage, almost 1000 genes are upregulated, which most likely indicates a slight ZGA wave [15]. As demonstrated by the ongoing degradation of the maternal X chromosome genes at the morula stage, the absolute counting of mRNA molecules reveals that the degradation of maternal mRNA begins at the first cleavage stage and continues to be incomplete at the eight-cell stage [3,12]. All three lineages’ mature cells can be easily recognized and described. According to their cellular functions, the EPI-specific, PrE-specific, and TE-specific genes enrich for genes related to apical plasma membrane and active transmembrane transporter activity, morphogenesis of epithelium and endoderm development, and stem cell maintenance and cell fate specification, respectively [3]. *NANOG* and *SOX2* for EPI, *GATA4* and *PDGFRA* for PrE, and *GATA2* and *GATA3* for TE are among the lineage-specific marker genes that have been found. The early intermediate cells and the developmental path of these three lineages have also been investigated in these investigations. According to the results, lineage-specific genes exhibit limited co-expression in the early intermediate cells [3,13]. The scRNA-seq data demonstrate that the human ICM and TE lineages, as well as the EPI and PrE lineages, are distinct at embryonic day 5 (post-fertilization, corresponding to the early blastocyst stage). This leads to a scenario of contemporaneous establishment of three lineages coincident with blastocyst formation for human preimplantation development, which is different from the mouse model [3]. However, another work supports this hypothesis by combining three human preimplantation scRNA-seq datasets for analysis [13]. These datasets are from [11,15]. An unsolved issue is the XCI in human preimplantation embryos [3,11,15].

To obtain dosage compensation of X chromosome transcripts, the paternal X chromosome is epigenetically repressed as imprinted XCI in the mouse preimplantation embryo [12]. On the other hand, imprinted XCI is not performed on human preimplantation embryos [16]. The scRNA-seq research has suggested a model of expression dampening of both X chromosomes during human preimplantation development [3]. However, a model of random XCI during the stage is supported by a subsequent analysis utilizing a novel pipeline of the same scRNA-seq dataset [17].

Zhou et al. [18] used single-cell multi-omics sequencing to analyze over 8000 cells from 65 embryos both before and after transplantation, simulating the implantation growth of fertilized eggs following artificial insemination in vitro. The findings offer vital information for the in vitro derivation and guided differentiation of pluripotent stem cells, as well as insights into the intricate molecular pathways that govern the implantation process of human embryos. Resources for a correct knowledge of human preimplantation development are provided by the scRNA-seq datasets. There are still many gaps in our understanding of the molecular mechanisms underlying human blastocyst formation. Nonetheless, there are numerous global limitations on access to and use of human embryos for research, which may restrict scientific advancements. Recent studies have sparked considerable interest with the development of stem cell-derived embryo models that resemble blastocyst-stage embryos, commonly referred to as blastoids.

## 5. Blastoids: Pre-Implantation Models

New developments in stem cell-derived embryo models, such as blastoids, vesicular structures that resemble blastocysts, open up new possibilities and offer scalable and comprehensive sources for in vitro embryogenesis modeling. The first mouse blastoids were successfully generated by Rivron et al. [19], setting a precedent. Using the self-organizing characteristic of the mammalian embryo, mouse blastoids are produced by aggregating several kinds of stem cells, such as pluripotent and extraembryonic stem cells, in non-adherent hydrogel microwells. A variety of morphogens that affect the specification of the TE and epiblast lineages are used to cultivate these cells. This process of self-assembly produces in vitro structures that are remarkably similar to blastocysts in vivo. In 2021, human pluripotent stem cells (hPSCs) were used to construct the first human blastoids, proving that they could form cell lineages and structures like those of human blastocysts (Figure 3) [20]. Prior studies have demonstrated that human blastoids can replicate all major cell lineages of human blastocysts, allowing for the study of blastocyst implantation and development. Another area of study that looks at cellular differentiation, division, and tissue formation throughout the early stages of human embryonic development is early development and stem cell research [21].

In order to generate blastoids in 4 days, a prior study cultivated hPSC and iPSC lines on PD0325901, XAV939, Gö6983, and LIF (PXGL) medium and verified that the three fundamental cell lineages—trophectoderm, EPI, and PrE—were generated in a manner akin to blastocyst formation. Additionally, transcriptome research showed that the cells’ gene expression patterns were highly similar, and it was established that human blastoids have a cellular organization resembling that of blastocysts [22]. For eight days, human ESC and iPSC lines were cultured in 5iLAF culture media to produce blastoids in a prior study [23], and confirmed that human SEMs recapitulate the lineage differentiation of the early human post-implantation embryo, according to single-cell transcriptome research. In previous studies, blastoids were produced in six days using human-naive induced PSCs (hniPSCs) lines and a mix of 5iLAF and PXGL culture media [24], and confirmed that the transcriptional characteristics of human early blastocysts and other models are replicated in spontaneous blastoids. In a prior study, hniPSC lines were cultivated for four days in a PXGL growth medium in order to produce blastoids [25]. This was obvious by the expression of essential trophectoderm markers such as *CDX2*, *GATA2*, or *GATA3* in the outer cell layer, enclosing an inner cellular aggregate expressing pluripotency markers such as *OCT4*, *NANOG*, or *SOX2*. The occurrence of hypoblast-like cells expressing markers such as *GATA6*, *GATA4*, or *SOX17* was also reported in the majority of these human blastoid models [26]. Additionally, single-cell transcriptome profiling and integrative analyses with known human blastocyst datasets verified the presence of epiblast, trophectoderm, and hypoblast-lineage cells within the human blastoids. Human blastoid models have enormous potential as a novel and adaptable technique to advance our knowledge of human development. Research on blastocysts and possible clinical translational studies resulting in better human reproductive technologies may be made possible by blastoid models. Although blastocyst lineages have been found in the majority of blastoid models by single-cell transcriptome analysis [26]. Research on blastoids has started to look into how signaling, including tissue crosstalk, plays an important role in determining cell fate. For instance, polar trophectoderm cells’ development is dependent on signals from epiblast cells, which enables the cells to adhere to the endometrium. Human blastoid derivation requires phosphorylinositol 3-kinase (PI3K)/AKT, mTOR, and AMPK pathways, while FGF/MAPK signaling is required for hypoblast development. In order to better understand the mechanisms of early human embryogenesis, future research will be able to examine cell fate decisions, morphogenesis, and epigenetic regulatory mechanisms such as X-chromosome inactivation, DNA methylation, and transposon biology. Because blastoids can also develop in human embryo culture settings, they could be utilized to enhance media formulations with potential therapeutic applications. Even though blastoids are a more ethical alternative, they are not a replacement for human embryos in research [27,28].

The phases that come before the blastocyst stage, notably the early cleavage and morula stages, which are particularly prone to failure, are not modeled by blastoids. While some blastoid models take longer and induce hypoblast cell specification prior to trophectoderm induction, others produce lineages in accordance with the rate and order of blastocyst development. Another drawback of blastoids and stem cell embryo models generally is that existing procedures are not entirely optimized and may result in the formation of cell types that should not be included in the model. Cells that are asynchronous with the regular developmental program, such as amnion-like and ExM-like cells found in certain blastoid models, are referred to as off-target cells [29,30].

## 6. Peri-Implantation Development

When a human embryo develops from day 8 to day 12, the mature blastocyst implants into the uterine wall. Usually, this time frame is referred to as peri-implantation. One of the most enigmatic developmental stages in mammalian embryogenesis is implantation. However, because of the limited access to and monitoring technology of the embryo early after implantation in vivo, the implantation of the whole conceptus into the maternal endometrium is less understood [31]. One of the main reasons for early pregnancy loss is implantation failure. It is estimated that between 40 and 60 percent of human conceptions end in failure, with implantation accounting for the majority of these losses. In order to uncover the fundamental processes guiding the implantation of human embryos, it is crucial to investigate the lineage specificity and associated patterns of the transcriptome and DNA methylome during implantation. In 2019, Tang and colleagues simulated human embryo implantation and used scCOOL-seq to investigate the gene-expression network and lineage-specific DNA methylation patterns of human peri-implantation embryos at single-cell resolution [18]. They examined over 8000 individual cells from 65 human peri-implantation embryos, performed single-cell whole-genome sequencing using the MALBAC amplification technique, built single-cell RNA-seq libraries using STRT-Seq, and analyzed the DNA methylome using single-cell whole-genome bisulfite sequencing (scBS-seq).

Additionally, they determined which particular genes are expressed in TE, PE, and EPI cells. Furthermore, they discovered that following implantation, the ratio of X chromosomes to autosomes in female cells was somewhat larger than that in male cells, based on the examination of parental allele-specific expression of X-chromosome-linked genes. They examined the lineage-specific DNA methylation dynamics during implantation using the single-cell Trio-seq2 strategy. They discovered that the PE lineage’s genome remethylated significantly more slowly than that of the EPI and TE lineages. This suggests that the embryos started DNA remethylation soon after the blastocyst stage and that PE, EPI, and TE displayed clearly different and asynchronous DNA remethylation patterns [32]. Similarly, in 2019, Yuan and colleagues defined the transcriptome landscape of placental trophoblast (TB) from cultivated human blastocysts during the implantation period using single-cell RNA sequencing techniques [32]. By analyzing the transcriptomes of cytoTB (CTB), syncytioTB (STB), and migratory TB (MTB) isolated from embryos at increasing days in culture, they provided insight into the processes causing early placental appearance. They discovered that these three types of TB cells show up in sequencing, with MTB exhibiting a mixed MTB/CTB phenotype emerging around D10, mature STB peaking at D10, and CTB with some STB characteristics emerging at D8.

Around D12, mature MTB begins to migrate away from the conceptus’ main body, CTB commences a new stage of proliferation, and STB production declines [32]. Using single-cell RNA sequencing, the other group recently identified the fundamental regulatory mechanism causing trophoblast destiny divergence during the human peri-implantation period [33]. T-box transcription factor 3 (TBX3) is crucial for mediating the differentiation of cytotrophoblast into syncytiotrophoblast, according to their analysis of the transcriptome of trophoblast cells using the SMART-seq2 assay [32]. Tan et al. used single-cell RNA sequencing to determine the underlying transcriptional landscape following ZIKV infection during the pre- and peri-implantation phase [34]. They stated that prenatal abnormalities and miscarriages were induced by ZIKV infection, which was linked to the loss of neural progenitor cells and trophectoderm [34].

## 7. Gastrulation

The beginning of the primitive streak and the subsequent development of primary germ layers, such as definitive endoderm (DE), mesoderm, and (neuro-) ectoderm, characterize the embryonic process of gastrulation. In order to guide mesodermal and/or endodermal progenitors on the inside of the embryo and the ectoderm on the outside, the germ layers coordinate cell division, movement, and rearrangement during the gastrulation process. This leads to destiny patterning. In vivo, the establishment of germ layer patterning is regulated in concert by various developmental signaling pathways, such as Nodal/TGFβ, BMP, WNT, and FGF. Tissue patterning, cell fate specification, and rearrangements are determined by these intricately linked signaling networks [18].

The early post-implantation human embryo investigation is very challenging, for both ethical and technical reasons, as in vivo studies are not possible, and human embryos only up to the blastocyst stage are typically cultured in vitro. Complex embryo models have also been created by combining human naive PSCs with extra-embryonic-like stem cells. In one study, assembloids were created by combining human naive PSCs with extra-embryonic cells, which serve as a signaling nest for the embryonic compartment. The authors used analyses of human embryos cultured in vitro to fine-tune the activation of key signaling pathways, including BMP, WNT, and NODAL/ACTIVIN-A. After self-organization without exogenous cues, a stem cell-based embryo model was formed, complete with an epiblast disk, amniotic and yolk sac-like cavities, PGCLCs, and ExM-like cells, which in this system are derived from epiblast-like cells. This is an excellent illustration of how embryo studies can be used to logically design protocols to enhance stem cell-based embryo models [35]. In another study, embryonic and hypoblast-like cells were combined to create a bilaminar structure made up of a pluripotent epiblast and an amnion-like epithelium encircled by cells that resembled the yolk sac. Using a transwell assay, TSCs were added to demonstrate the usefulness of this model for studying intercellular communication; TSCs secrete IL-6, which causes amniotic cavitation and epiblast proliferation. Finally, a model of the post-implantation human embryo was created by combining epiblast-, hypoblast-, ExM-, and trophoblast-like cells, which were derived from naive human PSCs. The resulting structures have an epiblast disk, amniotic and yolk sac-like cavities, ExM-like cells, and PGCLCs; additionally, they break the anterior–posterior symmetry and exhibit AVE and primitive streak-like domains [36]. The presence of an underdeveloped trophoblast and the incomplete spatial organization of the ExM are two limitations of this model, which also replicates important features of post-implantation human embryos. The protocol’s low efficiency will also need to be improved in order to support functional studies that break down the mechanisms of development. Three-dimensional aggregates of primed and enlarged potential human PSCs also build more complicated embryo models. Primed cells have been shown to produce structures with an epiblast disk, cavities resembling the amniotic and yolk sacs, PGCLCs, and ExM-like cells by first supplying a hypoblast induction medium and then an amnion induction medium. According to this hypothesis and pseudotime analysis, the hypoblast produces ExM-like cells, while the posterior epiblast cells produce PGCLCs. Although the capacity of primed PSCs to differentiate into cells that resemble hypoblast has not been functionally assessed. Two or more cell types have also been combined to model gastrulation; these cell types must self-assemble to recapitulate intercellular interactions and produce an embryo-like morphology. To demonstrate that yolk sac-like cells prevent PSCs from differentiating into mesoderm and endoderm by blocking BMP and WNT signaling, they were combined with PSCs in a 3D gel. However, no embryo-like architecture was seen, which was resolved by co-culturing human PSCs, which are representative of the post-implantation epiblast, and GATA6-over-expressing PSCs, which mimic the yolk sac, in 2D [7,24]. Trophoblast Organoid (TO) offers a helpful model for examining placental development in humans. In order to give technique and new research cues for future studies of human placentation, Bai-Mei Zhuang et al. described a single-cell atlas for TO and its applications on EVT differentiation and communications with decidual natural killer cells (dNKs) [37]. Shannon et al. create a single-cell transcriptome atlas of two trophoblast organoids that completely characterize the similarities and disparities in connection to trophoblasts from the placental-maternal interface [38]. Currently, TOs can be produced from blastocyst cells, TSCs, or primary cells. This gives researchers more freedom to choose the access method that best suits their requirements. A fascinating alternative method for analyzing trophoblast formation and differentiation, which is thought to be the cause of many pregnancy disorders, TOs have demonstrated the ability to capture the complexities of in vivo first-trimester placentation. Our knowledge of trophoblast cellular dynamics during the first trimester has significantly increased because of the work of scRNA-seq studies.

Cytotrophoblast (CTB), syncytiotrophoblast (SYN), and extravillous trophoblast (EVT) are the three primary cell types that can be differentiated from the trophoblast stem cell progenitor pool [39]. Although the broad categories of CTB, SYN, and EVT are correct, further nuances and distinctions based on spatial arrangement and protein expression/transcriptomics have helped to highlight the actual diversity of these cell types. Trophoblast organoid models may be able to simulate every stage of pregnancy and provide previously unheard-of access to the first trimester. New cell types that can be represented in vitro have been discovered as a result of the recent expansion of understanding regarding the dynamics of cells in the first trimester. Currently, the most sophisticated method for evaluating the accuracy of these models is single-cell transcriptomics (Figure 4). In the same year, Göttgens and colleagues created a genetic map of cellular differentiation from pluripotency toward endoderm and hematoendothelial lineages and presented the transcriptional profiles of 116,312 single cells from mouse embryos, with a median of 3436 genes found per cell. They also emphasized the crucial role that TAL1 plays in bloodline evolution [40]. Using single-cell RNA sequencing techniques, Moskowitz and colleagues in 2020 discovered that the Hedgehog (Hh)-fibroblast growth factor (FGF) signaling axis is necessary for the establishment of anterior mesoderm (AM) patterning during gastrulation [41]. They observed unique anterior mesoderm abnormalities in mouse embryo development by using transcriptional profiling and drop-seq to examine anterior–posterior axis patterning in a mesoderm-specific Hh pathway mutant. They came to the conclusion that Hh signaling has a previously unidentified role in the development of particular anterior mesoderm lineages by being necessary for FGF pathway activity in nascent mesoderm during gastrulation. Similarly to this, Arnold et al. (2021) used scRNA-seq, genetic destiny labeling, and imaging techniques to pinpoint the exact spatiotemporal pattern of AM and DE progenitor emergence during mouse embryo gastrulation germ layer development [42]. They discovered that AM and DE lineages were distinct from cells that expressed Eomes. Meissner and colleagues used scRNA-seq in 2020 to concurrently retrieve reliable transcriptional and morphological data from many mutant mouse embryos during the gastrulation stage [43]. They discovered that whereas PRC2 is dominant in germline restriction, PRC1 and PRC2 components exhibit significant cooperativity. The epigenetic modification, including cytosine methylation by DNA methyltransferases (DNMTs) and demethylation caused by oxidation of 5-methylcytosine by the Ten-eleven translocation (Tet) family, was analyzed by Sun and colleagues in 2016 using scRNA-seq, conventional bisulfite sequencing, and Tet-assisted bisulfite sequencing (TAB-seq) technologies. They found that TET-mediated demethylation and methylation regulate Lefty-Nodal signaling to dictate primitive streak patterning during mouse embryo gastrulation [44]. An in vitro culture technique was also developed by Wang and colleagues in 2019 to aid in the cynomolgus monkey embryo’s development past early gastrulation [45]. They investigated the traits and processes influencing lineage specification during embryo post-implantation, and they recreated the segregation of epiblasts and hypoblasts, the creation of the anterior–posterior axis, and the appearance of the primordial germ cells. To replicate developmental milestones and 3D architectures of the embryonic disk, amnion, and production of primitive streak anlage, Li and colleagues created a 3D in vitro human blastocyst-culture system in 2020 [4]. Using single-cell transcriptome profiling, they identified the regulatory network governing the segregation of trophoblast, primitive endoderm, and epiblast. However, relying solely on single-cell transcriptomics is inadequate, even if certain aspects of the in vitro model exhibit gene expression patterns that are comparable to those of its in vivo equivalent.

## 8. Neurulation

Neurulation, which is necessary for the neural tube’s creation and the central nervous system’s subsequent development, starts right after the embryo gastrulates. The morphogens released from the surrounding tissues and the biophysical signals from the extracellular matrix directly control the folding and fusion of the neural plate to create the neural tube [57]. In order to mediate the activation of genetic networks, including the activation of transcriptional factors, DNA methylation, and demethylation, these extracellular inductive cues are sensed and transduced through intracellular signaling pathways. This process directs the fate specification of progenitor cells within the neural tube. Brivanlou and colleagues recently used micropattern technology to create a neuruloid structure that replicates early human neurulation [58]. Then, using scRNA-seq technology, they discovered the exact identities and time of fate specification during neurulation. In two separate investigations, they evaluated more than 5105 single cells, discovered more than 100 genes with a pattern of expression unique to each population, and revealed landmarks of neural, neural crest, sensory placode, and epidermal gene expression.

By examining scRNA-seq data of neuruloids in wild type and 56 CAG background, they also deciphered the molecular mechanisms underlying the morphogenetic defect of Huntington’s disease (HD). They discovered that the WNT/PCP signaling pathway was downregulated, cytoskeleton-associated gene expression was drastically reduced, and actin-myosin contraction was significantly reduced [58]. In 2018, Xu and colleagues used scRNA-seq and ATAC-seq to examine the genome-wide transcriptome profile of individual cells in order to identify subpopulation specification and cell fate decision during human neurulation [59]. They analyzed the dynamics of chromatin accessibility across the stages of brain differentiation and discovered probable new transcription factors. All of these studies work together to advance our knowledge of the fundamental processes driving neuronal illness, cell fate specification, and neurulation.

## 9. Organogenesis

Trapnell and Shendure’s groups developed a high-throughput sequencing method called single-cell combinatorial-indexing RNA-sequencing analysis (sci-RNA-seqs) and analyzed over 2 million cells from various organogenesis-stage mouse embryos and recovered a median of 671 UMIs [2]. After gastrulation, three germ layers continue to develop into different organs, and during organogenesis, embryos grow from hundreds of cells to millions of cells. scRNA-seq provides a powerful strategy to map the transcriptional landscape at the single-cell level during early organ development. They discovered that several trajectories, like branching paths and the ability of some cell types to arise from many origins, are more complex than a straightforward linear journey, underscoring the intricacy of organogenesis [2].

In order to investigate the evolutionary and developmental relationships among different organs and cell types during mouse organogenesis, Tang and colleagues (2018) used scRNA-seq technology to analyze the transcriptomic features of approximately 2000 individual cells from eight organs and tissues from seven mouse embryos between E9.5 and E11.5 [60]. They mapped gene regulatory networks, created regulon matrices, and carried out hierarchy clustering using the SCENIC algorithm. Using hematopoietic, neural, mesodermal, and epithelial characteristics, they were able to form four main groups. They also discovered a hybrid epithelial/mesenchymal (E/M) condition in epithelial cells, indicating that endodermal organs frequently undergo this transition.

Using scRNA-seq, another study from Göttgen’s lab described how the leukotriene pathway mediates the development of blood progenitors during mouse organogenesis [61]. The transcriptome landscape of over 20,000 individual cells from mouse embryos was characterized during the transition from gastrulation to organogenesis. In the ordering of somatic progenitor cells, they discovered about 20 major cell types, as well as dynamic transcription waves and potential regulators. Additionally, by examining the whole transcriptomes of the hemogenic endothelium cells and the blood progenitors, they were able to determine the role of the leukotriene biosynthesis pathway as a regulator in modulating early blood development [61].

Maehr and colleagues examined eight days of thymus organogenesis in humans using scRNA-sequencing based on the drop-seq technology. They identified the cell-specific expression patterns in stromal and blood populations, profiled over 25,000 cells, exposed cellular heterogeneity, and investigated developmental dynamics. They discovered that embryonic thymus-resident cells may be involved in the etiologies of autoimmune diseases by merging the data from genome-wide association studies and genes linked to autoimmune diseases by the cell atlas [62].

Similarly, Hu and colleagues used droplet-based and well-based (STRT-seq) methods in 2019 to map the transcriptional landscape of human early T lymphopoiesis from several hemogenic and hematopoietic locations spanning embryonic and fetal stages [63]. They discovered a new group of prethymic lymphoid progenitors in the aorta-gonad-mesonephros region and a subtype of early thymic progenitors that possessed characteristics in common with a subset of lymphoid progenitors in the fetal liver. These studies offered valuable insights into human T lymphocyte regeneration and early T lymphopoiesis. The recent description and spatially resolved single-cell analysis of an intact gastrulation-stage human embryo by Tyser et al., 2021 [46], suggests that additional insights into embryonic patterning will be gained from further direct analysis of human post-gastrulation embryos. Since no experimental interventions in intact embryos beyond gastrulation are practical or acceptable, stem cell-derived models of these developmental stages, such as the 3D micropatterns and polarized gastruloids, become increasingly important for recapitulating the developmental attributes of gastrulation and tissue patterning.

The primary advantage of peri-gastruloids over previously known gastruloids is the quantity of hypoblasts generated, which offer crucial signals for fetal patterning. The researchers discovered through single-cell RNA sequencing that these structures give rise to cell lineages that, at the single-cell level, exhibit striking transcriptome similarities between peri-gastruloids and primate embryos. This cutting-edge peri-gastruloid platform offers exciting opportunities for additional research beyond the gastrulation stage and has enormous potential for using physiological developmental processes to engineer human tissue, which could have applications in regenerative medicine [64].

Gastruloids are collections of a particular number of ES cells that, in specific culture circumstances, go through symmetry breaking, controlled proliferation, and the specification of all three germ layers that are typical of vertebrate embryos and their offspring. The posterior region of gastruloids exhibits a variety of cell types and gene expression organization that is similar to that of the embryo, according to single-cell analysis and spatial transcriptomics [65,66,67,68]. In the extending posterior region, derivatives of the three germ layers, but especially the various classes of mesoderm, are arranged along the AP and DV axes, with an even bilateral asymmetry across a distinct midline. Due to the lack of neural cells at the front end, somitogenesis extends to the most anterior limit, where the heart primordium is also located. Furthermore, Tbx1, a marker of the second heart field that also contains progenitors for the cranial mesoderm, exhibits a crescent of expression in the anterior region during the early stages of gastruloid development [66]. Both the presence of cardiopharyngeal mesoderm and a population of neural crest cells that tomo-sequencing assigns to the anterior portion of the gastruloid are confirmed by single-cell analysis. Additionally, patches of genes defining placodal territory can be found in these places [66,67].

## 10. Developmental Trajectories, Heterogeneity, and Transcriptomic Signatures

Single-cell RNA sequencing (scRNA-seq) has revolutionized our understanding of early human development by enabling high-resolution mapping of gene expression at the individual cell level. The molecular markers, lineage segregation patterns, and dynamic trajectories that underpin the earliest phases of human development—specifically, from the zygote to the blastocyst and the early post-implantation stages—have been made visible by this approach. Yan et al. (2013) pioneered the use of single-cell RNA sequencing (scRNA-seq) on human preimplantation embryos. They profiled 124 distinct cells, including human embryonic stem cells (hESCs) and metaphase II oocytes to late blastocysts. This landmark research determined that zygotic genome activation (ZGA) occurs between the four-cell and eight-cell phases, with important ZGA genes like *DPPA3*, *ZSCAN4*, and *EIF1AX* indicating the transcriptional switch from mother to embryo. Yan et al. went on to describe lineage-specific transcriptome signatures, finding that trophectoderm (TE) cells expressed *CDX2* and *GATA3*, while pluripotency factors such as *NANOG*, *SOX2*, and *POU5F1* were enriched in the inner cell mass (ICM) [15].

By analyzing individual cells from human blastocysts, Blakeley et al. (2015) expanded on this and clarified our understanding of lineage segregation by identifying the three main lineages: trophectoderm (TE), primitive endoderm (PrE), and epiblast (EPI). Strong lineage-specific markers were found, including *CDX2* and *GATA3* for TE, *GATA4* and *SOX17* for PrE, and *NANOG* and *SOX2* for EPI. Additionally, their data demonstrated the role of signaling pathways such as FGF/MAPK, which is especially enriched in PrE. Notable species differences indicate that human embryos undergo lineage segregation earlier and more discretely than mouse embryos [11].

Petropoulos et al. (2016) extended these results by conducting high-throughput scRNA-seq on 1529 cells from 88 embryos spanning the eight-cell to late blastocyst stages. Their findings highlighted significant transcriptional variability within lineages along with gradual and asynchronous lineage segregation. While *GATA6* and *SOX17* identified primitive endoderm, *GATA3*, *CDX2*, and *KRT18* identified trophectoderm, important lineage markers like *NANOG*, *SOX2*, and *KLF17* designated epiblast cells. Crucially, their analysis identified signaling pathways, including FGF, TGF-β, and WNT, as key modulators of differentiation and lineage commitment [3].

Xiang et al. (2020), further advancing this, used scRNA-seq on 3D-cultured human embryos up to day 14, identifying new post-implantation lineages such as amniotic epithelium (*TFAP2A*, *ISL1*), yolk sac endoderm (*SOX17*, *AFP*), and mesoderm (*TBXT*, *MESP1*). Their research revealed important developmental events, such as the epithelial-to-mesenchymal transition (EMT), which is controlled by the BMP, NODAL, and WNT pathways during the creation of primitive streaks [4].

Importantly, computational technologies like RNA velocity (Bergen et al., 2020) [69] and pseudotime analysis (Monocle; Trapnell et al., 2014) [70] have made it possible to forecast future cell states and reconstruct developmental trajectories, going beyond static gene expression snapshots. These techniques determine bifurcation sites during lineage commitment and infer the temporal ordering of individual cells along differentiation pathways. Trajectory inference, for example, demonstrated the continuous and asynchronous migration of cells from undifferentiated states toward TE, EPI, or PrE lineages in the dataset of Petropoulos et al. Similarly, by predicting the directionality of cell fate transitions and shedding light on transient intermediate states, RNA velocity techniques applied to peri-implantation datasets have captured dynamic events that are otherwise impossible to study in human embryos due to ethical constraints. The intricate interactions that control early human embryogenesis between transcriptome heterogeneity, lineage-specific markers, and dynamic developmental trajectories have been clarified by these investigations taken together. Our knowledge of how pluripotent and multipotent cells arise, differentiate, and commit to specific fates via strictly controlled gene expression programs and signaling networks is enhanced by this integrated molecular framework.

In addition to producing a reference map of lineage-specific gene expression during early human development, these landmark single-cell investigations (e.g., Petropoulos et al., 2016; Xiang et al., 2020) also demonstrated dynamic and diverse transcriptional programs across embryonic lineages [3,4]. Together, these investigations have yielded important information about the timing, flexibility, and regulation of cell fate decisions, in addition to a list of markers specific to a given lineage.

First, they discovered a slow and asynchronous model of lineage commitment, which contradicted the traditional theory of synchronized and discrete lineage segregation. This implies that rather than occurring at clearly defined stages, decisions about cell destiny occur over a continuum of transitional states.

Second, these investigations presented a new interpretation of pluripotency as a range of transcriptional states that change dynamically during implantation and gastrulation rather than as a binary on/off state. This has important ramifications for how we create in vitro models and understand early human development.

Third, the combination of RNA velocity and trajectory inference in later research (e.g., Tyser et al., 2021) has shed light on the directionality of fate transitions and the temporal evolution of cell states [46]. These results demonstrated that the pathways cells take toward particular lineages are determined by transcriptional dynamics rather than static snapshots of gene expression. Together, these revelations mark conceptual breakthroughs that transform our knowledge of human embryogenesis from static lineage blueprints to probabilistic and dynamic developmental pathways.

## 11. Technical Limitations and Challenges of Single-Cell RNA Sequencing in Human Embryo Studies

The studies of early human development have greatly benefited from single-cell RNA sequencing (scRNA-seq); yet, there are still a number of technical restrictions and difficulties, especially when working with human embryos.

### 11.1. Low Input Material and Ethical Restrictions

The scarcity of human embryos for study, which results from stringent ethical guidelines and societal concerns, is one of the main obstacles. Sample sizes are limited by this scarcity, which lowers statistical power and makes it more difficult to identify uncommon cell populations. Technical problems like RNA degradation and stochastic gene dropout are also due to the small quantity of starting material in embryos. These problems might bias gene expression profiles and mask low-abundance transcripts that are essential for comprehending delicate developmental processes (Petropoulos et al., 2016; Yan et al., 2013; Ziegenhain et al., 2017) [3,15,71].

### 11.2. Limitations in Capturing Spatial Information and Transient Cell States

In order to interpret how cell position and microenvironmental cues impact lineage specification and cellular interactions, traditional scRNA-seq methods necessitate the dissociation of embryonic tissues into single cells, which results in the loss of spatial context. While this gap has been partially filled by new spatial transcriptomics technologies (Stahl et al., 2016; Asp et al., 2019) [72,73], existing methods frequently lack the resolution or throughput necessary to completely capture the intricate three-dimensional architecture of early embryos. Furthermore, some developmental cell states are fast and ephemeral, which makes detection difficult. To capture these fleeting states, sensitive techniques that can differentiate them from more stable populations are needed (La Manno et al., 2018; Bergen et al., 2020) [69,74].

### 11.3. Integration Challenges with Other Omics Data for Holistic Understanding

It is crucial to integrate scRNA-seq data with other omics layers, including proteomics, metabolomics, and epigenomics, in order to completely understand the molecular pathways underpinning human embryogenesis. Nevertheless, there are computational and experimental challenges associated with multi-omics integration. The alignment and collaborative analysis of disparate data sources are complicated by their differing resolutions, sizes, and noise characteristics. Furthermore, cross-validation and model resilience are limited by the lack of complete datasets from the same samples. Advances in bioinformatics frameworks and experimental design are necessary to address these issues and provide comprehensive, systems-level insights into human development (Argelaguet et al., 2018; Ma et al., 2020) [75,76]. The potential of single-cell technologies in developmental biology and transferring research discoveries into clinical and therapeutic contexts depends on overcoming these technical constraints and difficulties.

## 12. Future Perspective

The process of early embryonic development is intricate and tightly controlled. Currently, the following issues affect embryonic development: In the early phases of embryonic development, there are not enough cells to systematically take advantage of these properties. Heterogeneity among cells is evident during the blastocyst stage. It is still unclear how cell differentiation is regulated during somatic embryogenesis. Furthermore, the study of embryo development depends on these important questions. These problems have been addressed in light of the quick development of scRNA-seq. Single-cell transcriptomics methods have become indispensable tools for analyzing cellular heterogeneity in individual tissues over the last ten years. It is anticipated that rapid technological advancements will increase the scope and complexity of scRNA-seq applications. In order to provide a systematic framework for understanding the molecular characteristics of cell types or states, cellular trajectories, molecular mechanisms of development and differentiation, and regulatory interactions between cells, comprehensive transcriptomic reference maps of all cell types in the bodies of diverse organisms, including humans, are being constructed.

Numerous fields have been greatly impacted by scRNA-seq, and the following objectives can be accomplished using this methodology: Cell lineages are traced; the variety of normal tissues and organs is revealed; new cell types and marker genes are found; and developmental differentiation mechanisms and gene expression traits in various cells from small samples are identified. Dissecting the variability in previously hidden cell populations is made easier by examining the biological traits of the main cell types of essential organs and tissues at various stages of embryonic development at the single-cell level. The ongoing development of single-cell sequencing methods has yielded many novel insights and significantly advanced developmental biology in recent years. Furthering our understanding of organ development, these approaches are also helpful in identifying the spatiotemporal specific activation properties of important signaling pathways and the intricate signal interaction mechanisms throughout organ development.

We will also be able to diagnose disorders more accurately thanks to the advancements in single-cell sequencing techniques in the field of embryonic development. This technology offers a theoretical foundation for the prevention and treatment of conditions with significant clinical implications, including congenital kidney disease, neurological and immunological deficiencies, pediatric leukemia, and congenital heart disease. The developmental abnormalities of human infertility, which affect roughly 10% to 15% of couples of reproductive age, will be better understood molecularly. We anticipate that this will be accomplished in the next years, which will ultimately result in improved disease diagnosis and treatment.

## Figures and Tables

**Figure 1 ijms-26-07741-f001:**
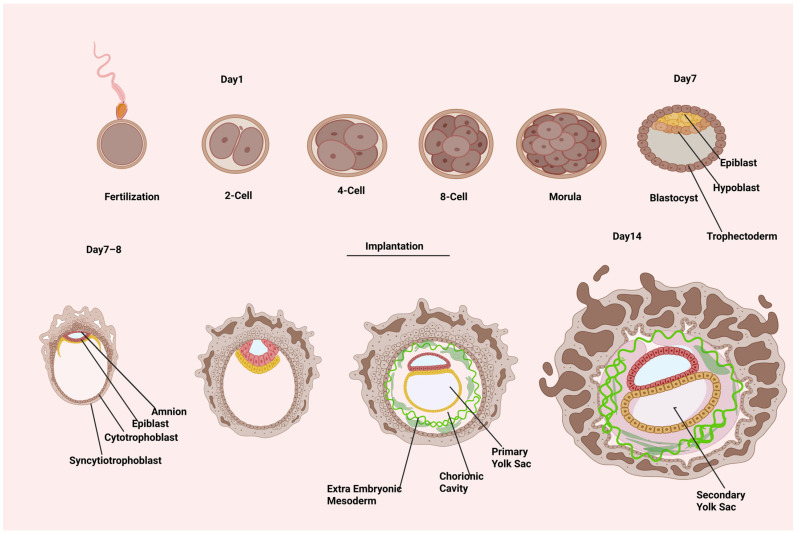
Developmental milestones in human embryo development.

**Figure 2 ijms-26-07741-f002:**
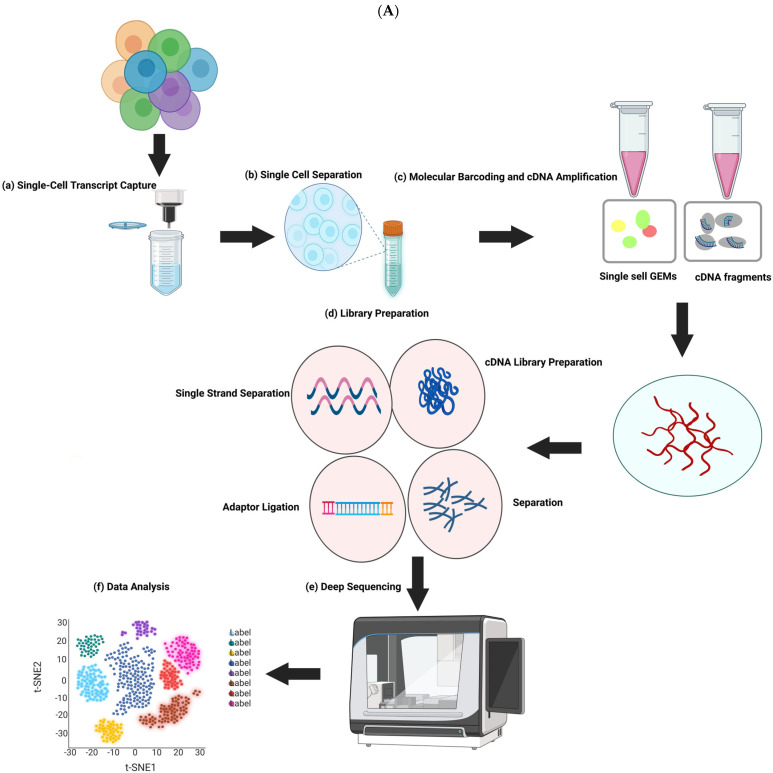
An overview of scRNA technology: (**A**) procedure of scRNA; (**B**) summary of scRNA analyses.

**Figure 3 ijms-26-07741-f003:**
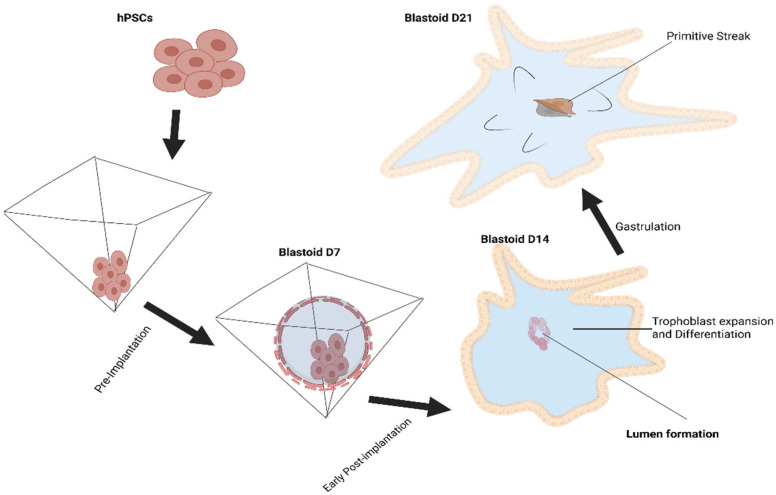
An overview of blastoid development.

**Figure 4 ijms-26-07741-f004:**
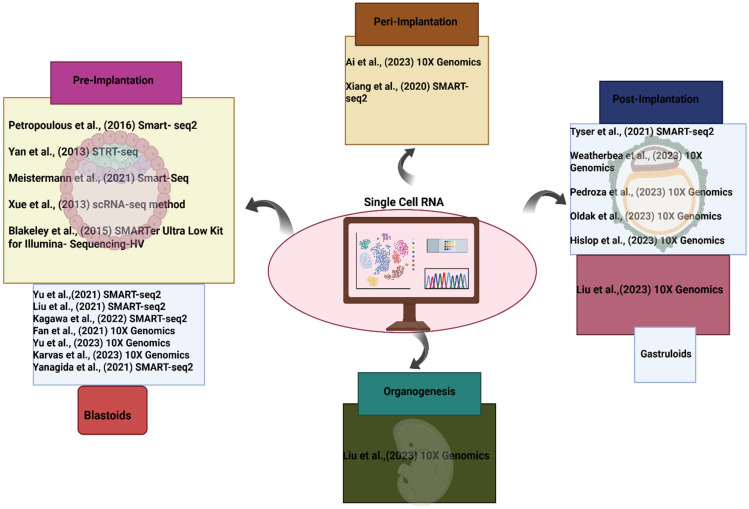
Summary of scRNA applications in human embryo development [3,4,11,14,15,20,22,23,35,46,47,48,49,50,51,52,53,54,55,56].

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
