# Peer review of "Single Cell RNA Sequencing and Its Impact on Understanding Human Embryo Development"

_ijms, 2025, doi:10.3390/ijms26167741_

Round 1

Reviewer 1 Report

Comments and Suggestions for Authors

This review summarizes single-cell RNA sequencing studies on human embryo development, highlighting the knowledge and insights gained through this transformative technology. The authors provide an overview of the field, discussing how single-cell approaches have advanced our understanding of the genome, transcriptome, and epigenome at the level of individual cells during early human development. Given the highly complex nature of human embryogenesis, and the ethical and technical limitations historically hindering its study, single-cell technologies indeed represent a significant breakthrough, enabling systematic characterization of cellular states and lineages.

Overall, the review is well-motivated, the references are reasonably cited, and the writing is generally clear. However, the manuscript currently reads more like a summary of existing studies rather than a critical review that synthesizes and interprets the findings in depth. To strengthen the impact and usefulness of this review for the field, I suggest the following:

  1. Given the focus on single-cell RNA sequencing, it is important that the review emphasizes the single-cell heterogeneity and transcriptome signatures during the early human development process that were learned from the single cell sequencing study. For example, key transcriptomic signatures that define specific cell types, states, or lineages during pre- and peri-implantation stages; How these signatures inform our understanding of developmental trajectories, cell fate decisions, and the molecular basis of early human development.
  2. Beyond summarizing individual studies, the authors should discuss what conceptual advances have emerged (e.g. in lineage specification, pluripotency dynamics, implantation stages), and provide deeper critical analysis and insights.
  3. Discuss technical limitations and challenges. Consider expand on current limitations of single-cell RNA-seq in studying human embryos, such as:
    • Low input material and ethical restrictions.
    • Limitations in capturing spatial information and transient cell states.
    • Integration challenges with other omics data for holistic understanding.

4. It would also be valuable to add the authors’ own views on:

    • Emerging technologies (e.g. spatial transcriptomics, single-cell multi-omics, lineage tracing).
    • Remaining key questions in human embryology that single-cell approaches could address in the near future.

5. Including more figures summarizing the new knowledge provided by single cell research could also significantly enhance the impact of this manuscript.

In summary, while the review provides a solid starting point, greater emphasis on critical discussion, authors’ perspectives, and synthesis of findings would make it a more valuable resource for readers in the developmental biology and genomics communities.

Author Response

Dear Reviewer

Thank you for giving us the opportunity to improve and resubmit our manuscript titled “Single cell RNA sequencing and its impact on understanding human embryo development”. We have carefully studied the comments and suggestions, and then added the corresponding information. We hope that the revision could be acceptable. Below is a point-by-point response to the reviewer’ comments.

  1. Given the focus on single-cell RNA sequencing, it is important that the review emphasizes the single-cell heterogeneity and transcriptome signatures during the early human development process that were learned from the single cell sequencing study. For example, key transcriptomic signatures that define specific cell types, states, or lineages during pre- and peri-implantation stages; How these signatures inform our understanding of developmental trajectories, cell fate decisions, and the molecular basis of early human development.

Response.1: Thank you so much for your comments, we added from line 526-579, it is clearly explained:

Exploring Early Human Development via Single-Cell RNA Sequencing: Developmental Trajectories, Heterogeneity, and Transcriptomic Signatures

Single-cell RNA sequencing (scRNA-seq) has revolutionized our understanding of early human development by enabling high-resolution mapping of gene expression at the individual cell level. The molecular markers, lineage segregation patterns, and dynamic trajectories that underpin the earliest phases of human development specifically, from the zygote to the blastocyst and the early post-implantation stages have been made visible by this approach. Yan et al. (2013) pioneer to employ single-cell RNA sequencing (scRNA-seq) on human preimplantation embryos. They profiled 124 distinct cells, including human embryonic stem cells (hESCs) and metaphase II oocytes to late blastocysts. This landmark research determined that zygotic genome activation (ZGA) occurs between the 4-cell and 8-cell phases, with important ZGA genes like DPPA3, ZSCAN4, and EIF1AX indicating the transcriptional switch from mother to embryo. Yan et al. went on to describe lineage-specific transcriptome signatures, finding that trophectoderm (TE) cells expressed CDX2 and GATA3, while pluripotency factors such as NANOG, SOX2, and POU5F1 were enriched in the inner cell mass (ICM).

By analyzing individual cells from human blastocysts, Blakeley et al. (2015) extended on this and clarified our understanding of lineage segregation by identifying the three main lineages: trophectoderm (TE), primitive endoderm (PrE), and epiblast (EPI). Strong lineage-specific markers were found, including CDX2 and GATA3 for TE, GATA4 and SOX17 for PrE, and NANOG and SOX2 for EPI. Additionally, their data demonstrated the role of signaling pathways such as FGF/MAPK, which is especially enriched in PrE. Notable species differences indicate that human embryos undergo lineage segregation earlier and more discretely than mouse embryos.

Petropoulos et al. (2016) extended these results by conducting high-throughput scRNA-seq on 1,529 cells from 88 embryos spanning the 8-cell to late blastocyst stages. Their findings highlighted significant transcriptional variability within lineages along with gradual and asynchronous lineage segregation. While GATA6 and SOX17 identified primitive endoderm, GATA3, CDX2, and KRT18 identified trophectoderm, while important lineage markers like NANOG, SOX2, and KLF17 designated epiblast cells. Crucially, their analysis identified signaling pathways including as FGF, TGF-β, and WNT as key modulators of differentiation and lineage commitment.

Xiang et al. (2020) further advancing this, used scRNA-seq on 3D-cultured human embryos up to day 14, identifying new post-implantation lineages such amniotic epithelium (TFAP2A, ISL1), yolk sac endoderm (SOX17, AFP), and mesoderm (TBXT, MESP1). Their research revealed important developmental events, such as the epithelial-to-mesenchymal transition (EMT), which is controlled by the BMP, NODAL, and WNT pathways during the creation of primitive streaks.

Importantly, computational technologies like RNA velocity (Bergen et al., 2020) and pseudotime analysis (Monocle; Trapnell et al., 2014) have made it possible to forecast future cell states and reconstruct developmental trajectories, going beyond static gene expression snapshots. These techniques determine bifurcation sites during lineage commitment and infer the temporal ordering of individual cells along differentiation pathways. Trajectory inference, for example, demonstrated the continuous and asynchronous migration of cells from undifferentiated states toward TE, EPI, or PrE lineages in the dataset of Petropoulos et al. Similarly, by predicting the directionality of cell fate transitions and shedding light on transient intermediate states, RNA velocity techniques applied to peri-implantation datasets have captured dynamic events that are otherwise impossible to study in human embryos due to ethical constraints. The intricate interactions that control early human embryogenesis between transcriptome heterogeneity, lineage-specific markers, and dynamic developmental trajectories have been clarified by these investigations taken together. Our knowledge of how pluripotent and multipotent cells arise, differentiate, and commit to specific fates via strictly controlled gene expression programs and signaling networks is enhanced by this integrated molecular framework.

  1. Beyond summarizing individual studies, the authors should discuss what conceptual advances have emerged (e.g. in lineage specification, pluripotency dynamics, implantation stages), and provide deeper critical analysis and insights.

Response.2: Thank you so much for your comments, we added from line 580-602, it is clearly explained

In addition to producing a reference map of lineage-specific gene expression during early human development, these landmark single-cell investigations (e.g., Petropoulos et al., 2016; Xiang et al., 2020) also demonstrated dynamic and diverse transcriptional programs across embryonic lineages. Together, these investigations have yielded important information about the timing, flexibility, and regulation of cell fate decisions in addition to a list of markers specific to a given lineage.

First, they discovered a slow and asynchronous model of lineage commitment, which contradicted the traditional theory of synchronized and discrete lineage segregation. This implies that rather than occurring at clearly defined stages, decisions about cell destiny occur over a continuum of transitional states.

Second, these investigations presented a new interpretation of pluripotency as a range of transcriptional states that change dynamically during implantation and gastrulation rather than as a binary on/off state. This has important ramifications for how we create in vitro models and understand early human development.

Third, the combination of RNA velocity and trajectory inference in later research (e.g., Tyser et al., 2021) has shed light on the directionality of fate transitions and the temporal evolution of cell states. These results demonstrated that the pathways cells take toward particular lineages are determined by transcriptional dynamics rather than static snapshots of gene expression.

Together, these revelations mark conceptual breakthroughs that transform our knowledge of human embryogenesis from static lineage blueprints to probabilistic and dynamic developmental pathways.

  1. Discuss technical limitations and challenges. Consider expand on current limitations of single-cell RNA-seq in studying human embryos, such as:
    • Low input material and ethical restrictions.
    • Limitations in capturing spatial information and transient cell states.
    • Integration challenges with other omics data for holistic understanding.

Response.3: Thank you so much for your comments, we added from line 603-640, it is clearly explained

Technical Limitations and Challenges of Single-Cell RNA Sequencing in Human Embryo Studies

The study of early human development has greatly benefited from single-cell RNA sequencing (scRNA-seq); yet, there are still a number of technical restrictions and difficulties, especially when working with human embryos.

Low Input Material and Ethical Restrictions

The scarcity of human embryos for study, which results from stringent ethical guidelines and societal concerns, is one of the main obstacles. Sample sizes are limited by this scarcity, which lowers statistical power and makes it more difficult to identify uncommon cell populations. Technical problems like RNA degradation and stochastic gene dropout are also small quantity of starting material in embryos. These problems might bias gene expression profiles and mask low-abundance transcripts that are essential for comprehending delicate developmental processes (Petropoulos et al., 2016; Yan et al., 2013; Ziegenhain et al., 2017).

Limitations in Capturing Spatial Information and Transient Cell States

In order to interpret how cell position and microenvironmental cues impact lineage specification and cellular interactions, traditional scRNA-seq methods necessitate the dissociation of embryonic tissues into single cells, which results in the loss of spatial context. While this gap has been partially filled by new spatial transcriptomics technologies (Ståhl et al., 2016; Asp et al., 2020), existing methods frequently lack the resolution or throughput necessary to completely capture the intricate three-dimensional architecture of early embryos. Furthermore, some developmental cell states are fast and ephemeral, which makes detection difficult. To capture these fleeting states, sensitive techniques that can differentiate them from more stable populations are needed (La Manno et al., 2018; Bergen et al., 2020).

Integration Challenges with Other Omics Data for Holistic Understanding

It is crucial to integrate scRNA-seq data with other omics layers, including proteomics, metabolomics, and epigenomics, in order to completely understand the molecular pathways underpinning human embryogenesis. Nevertheless, there are computational and experimental challenges associated with multi-omics integration. The alignment and collaborative analysis of disparate data sources are complicated by their differing resolutions, sizes, and noise characteristics. Furthermore, cross-validation and model resilience are limited by the lack of complete datasets from the same samples. Advances in bioinformatics frameworks and experimental design are necessary to address these issues and provide comprehensive, systems-level insights into human development (Argelaguet et al., 2019; Ma et al., 2020). The potential of single-cell technologies in developmental biology and transferring research discoveries into clinical and therapeutic contexts depend on overcoming these technical constraints and difficulties.

Second Reviewer

Line 255-256: In fact, implantation failure is considered by numerous sources a consequence of numerical chromosomal errors in the early embryo. Can this phenomenon be revealed by the methods discussed by the authors?

Response.1: Thank you so much for your comments, it is clearly explained

Characterizing gene expression dynamics at the single-cell level, exposing cell types, states, and developmental trajectories, is the main goal of scRNA-seq in the study of human development. Standard scRNA-seq procedures are unable to directly detect numerical chromosomal defects (aneuploidies), despite the fact that scRNA-seq offers abundant transcriptome data.

Nevertheless, scRNA-seq data can be used to identify chromosomal abnormalities in individual cells using additional computational techniques, such as determining copy number variations (CNVs) or mosaic aneuploidies from transcriptome patterns. These methods have been effectively used to detect aneuploid cells in early embryos, which are frequently connected to unsuccessful implantation.

Griffiths, J. A., Scialdone, A., & Marioni, J. C. (2017).
Mosaic autosomal aneuploidies are detectable from single-cell RNA-seq data.
BMC Genomics, 18(1), 904.

 Vera-Rodriguez, M., Chavez, S. L., Rubio, C., & Simon, C. (2015).
Prediction model for aneuploidy in early human embryo development revealed by single-cell analysis.

Line 549: Why is human infertility ascribed to developmental abnormalities?

Response.2:

Infertility in this context refers to the inability to conceive a child, which may be caused by embryos that have developmental defects that hinder them from progressing beyond the early stages. Developmental defects are not the only reason of infertility; problems with gametes, fertilization, the uterine environment, or hormone imbalances can also be involved. However, embryonic developmental defects are a primary cause of early pregnancy loss and implantation failure, which greatly increases the risk of infertility.

Line 269-270: Better explanation is needed, since the ratio of X chromosomes to autosomes in female cells is by default twice larger than that in male cells.

Response.3: Thank you so much for your comments, it is clearly explained

The statement makes reference to Zhang et al.'s (2021) findings, which showed that female epiblast cells had a little greater X-to-autosome expression ratio than male cells after implantation. X-chromosome inactivation (XCI) typically balances gene dosage, even though male cells only have one X chromosome and female cells have two. However, this study demonstrated that XCI is incomplete in the early post-implantation stages and that certain X-linked genes evade silencing by using parental allele-specific expression. Consequently, X-chromosomal expression in relation to autosomes was slightly higher in female cells than in male cells.

Fig. 4: Peri- instead of pri-implantation.

Response.4:  Thank you so much for your comments, it is clearly explained CONFIRMED

Line 420: What does “after neural development” mean here?

CONFIRMED

References: The listing of authors should be uniform.

CONFIRMED

Line 9: “because of” should be used instead of “despite”

CONFIRMED

Line 10: “therefore” is not needed

CONFIRMED

Line 66, 74, 338: Epiblast and hypoblast should be in singular.

CONFIRMED

Lines 77-78: Missing intervals.

CONFIRMED

Line 224: Should be “resulting” instead of “results”

CONFIRMED

Line 234: Should be “such as” instead of “such”

CONFIRMED

Line 240: The comma should be before “notably”.

CONFIRMED

Line 359: Very long paragraphs, such as the one starting at this line, should be broken.

CONFIRMED

Reviewer 2 Report

Comments and Suggestions for Authors

The presented review is comprehensive and addresses interesting problems, but needs improvements before publication.

Fig. 1: The drawing needs a major revision. The top left oocyte has a nucleus instead of a metaphase II plate, as should be by the start of fertilization. ICM, referenced in the text, is not indicated. On day 7-8, colors are reversed, and the “epiblast” label seems to point at the hypoblast. On “Implantation”, the “extraembryonic mesoderm” label seems to point at the endoderm, and the chorionic cavity is labeled the same as the primary yolk sac.

The fate of the primary yolk sac is not clarified.

Line 158: The eutherian blastula is usually called blastocyst, as elsewhere in the text.

Line 255-256: In fact, implantation failure is considered by numerous sources a consequence of numerical chromosomal errors in the early embryo. Can this phenomenon be revealed by the methods discussed by the authors?

Line 269-270: Better explanation is needed, since the ratio of X chromosomes to autosomes in female cells is by default twice larger than that in male cells.

Fig. 4: Peri- instead of pri-implantation.

Line 420: What does “after neural development” mean here?

Line 549: Why is human infertility ascribed to developmental abnormalities?

References: The listing of authors should be uniform.

Comments on the Quality of English Language

The manuscript needs thorough reading and editing by a person fluent in English.

Line 9: “because of” should be used instead of “despite”

Line 10: “therefore” is not needed

Line 66, 74, 338: Epiblast and hypoblast should be in singular.

Lines 77-78: Missing intervals.

Line 224: Should be “resulting” instead of “results”

Line 234: Should be “such as” instead of “such”

Line 240: The comma should be before “notably”.

Line 359: Very long paragraphs, such as the one starting at this line, should be broken.

Author Response

(The authors gave the same response as above.)

Round 2

Reviewer 2 Report

Comments and Suggestions for Authors

I approve the manuscript in its revised form.